# Capturing the Complexity of COVID-19 Research: Trend Analysis in the First Two Years of the Pandemic Using a Bayesian Probabilistic Model and Machine Learning Tools

Javier De La Hoz-M [1,*], Susana Mendes [2], María José Fernández-Gómez [3] and Yolanda González Silva [4]

1   Facultad de Ingeniería, Universidad del Magdalena, Santa Marta 470004, Colombia
2   MARE/ARNET, School of Tourism and Maritime Technology, Polytechnic of Leiria, 2520-614 Peniche, Portugal
3   Institute of Biomedical Research of Salamanca, 37008 Salamanca, Spain
4   Centro de Salud Ponferrada III, Gerencia de Asistencia Sanitaria del Bierzo (GASBI), 24403 Ponferrada, Spain
*   Correspondence: jdelahoz@unimagdalena.edu.co

**Abstract:** Publications about COVID-19 have occurred practically since the first outbreak. Therefore, studying the evolution of the scientific publications on COVID-19 can provide us with information on current research trends and can help researchers and policymakers to form a structured view of the existing evidence base of COVID-19 and provide new research directions. This growth rate was so impressive that the need for updated information and research tools become essential to mitigate the spread of the virus. Therefore, traditional bibliographic research procedures, such as systematic reviews and meta-analyses, become time-consuming and limited in focus. This study aims to study the scientific literature on COVID-19 that has been published since its inception and to map the evolution of research in the time range between February 2020 and January 2022. The search was carried out in PubMed extracting topics using text mining and latent Dirichlet allocation modeling and a trend analysis was performed to analyze the temporal variations in research for each topic. We also study the distribution of these topics between countries and journals. 126,334 peer-reviewed articles and 16 research topics were identified. The countries with the highest number of scientific publications were the United States of America, China, Italy, United Kingdom, and India, respectively. Regarding the distribution of the number of publications by journal, we found that of the 7040 sources *Int. J. Environ. Res. Public Health*, *PLoS ONE*, and *Sci. Rep.*, were the ones that led the publications on COVID-19. We discovered a growing tendency for eight topics (Prevention, Telemedicine, Vaccine immunity, Machine learning, Academic parameters, Risk factors and morbidity and mortality, Information synthesis methods, and Mental health), a falling trend for five of them (Epidemiology, COVID-19 pathology complications, Diagnostic test, Etiopathogenesis, and Political and health factors), and the rest varied throughout time with no discernible patterns (Therapeutics, Pharmacological and therapeutic target, and Repercussion health services).

**Keywords:** COVID-19; topic modeling; latent Dirichlet allocation; machine learning; text mining



## 1. Introduction

In March 2020, the World Health Organization declared the coronavirus outbreak a pandemic [1]. Since then, given the novelty of the disease, the scientific community has mobilized rapidly, reaching a considerably high number of scientific publications. As a result of the above, monitoring the rising database in medicine is becoming increasingly difficult, rendering traditional standard procedures such as systematic reviews and meta-analyses inappropriate approaches in an area as dynamic as the novel coronavirus [2]. Given the large number of publications, an approach that is more direct and has a broader reach is required.

Larsen and von Ins [3] stated that the worldwide increase in scientific literature can lead to researchers feeling overwhelmed and, therefore, their ability to carry out a review and follow-up of new research is effectively decimated.

Several comprehensive studies have been published on various aspects of the pandemic, including symptoms, treatments, and comorbidities [4–6]. Bibliometric analysis of studies on the COVID-19 pandemic has also been carried out [7–10]. However, the majority of the research looked at papers that were published during the first months of the COVID-19 pandemic being declared. As a result, several papers released since then have yet to be examined.

Our goals were to analyze the available scientific literature on COVID-19, identify the research topic, and describe the evolution of COVID-19 research to date, using a machine learning-based methodology. The significant worries of society about various facets of the pandemic's effects make scientific knowledge synthesis more vital than ever. Given the growing diversity of research topics related to COVID-19, quantitative studies are needed to better understand and answer the following concerns:

- Question 1 (Q1): What were the key publishing sources and major contributions to COVID-19 research?
- Question 2 (Q2): What are the major research topics in this field?
- Question 3 (Q3): How do these research topics evolve with time?
- Question 4 (Q4): What are the distributions of these topics across countries and journals?

## 2. Materials and Methods

### 2.1. Data Collection

Interventional Searching was conducted on 15 February 2022, using PubMed E-utilities using the following query: "COVID-19 (Title/Abstract) AND English (LA) AND Journal Article (PT] AND 2020/02/01 (dp]: 2022/01/31 (dp]". The illness COVID-19, rather than the virus, was the focus of this research. As a result, alternative search phrases or concepts were ignored in this inquiry. For each article, we obtained the title, keywords, abstract, date of publication, list of author affiliations, journal name, and PubMed identification number.

We regarded the country of affiliation of the first author to be the nation of origin of the article. If a nation's name was not contained in the affiliation, we utilized the most recently mentioned geographic entity and manually connected it to a country; for example, "Bogota" was linked to "Colombia".

We used bibliometric analysis to answer Q1. This enables the sample of publications to be used to determine various elements of scientific production [11,12]. In this section of the investigation, data were processed using bibliometrix [11], an open-source software written in the R programming language [13].

### 2.2. Data Preprocessing

Preprocessing is the first step in text mining techniques and their application, playing a crucial role in the entire procedure [14]. To increase the coherence of the topics, each abstract was tokenized using bigrams which are the combination of consecutive unigrams. Although preprocessing seems trivial, since the text is downloaded to the computer as a readable format, it must be converted to lowercase and punctuation marks, dashes, brackets, numbers, space blanks and other characters removed. In addition, a standard list of words called "stopword" was identified and eliminated, since their main function is to make a sentence grammatically correct (i.e., articles and prepositions).

Data preprocessing was carried out using the web-based tool LDAShiny [15], a package to R programming language [13]. As a result of these operations, a document term matrix was created (dtm).

### 2.3. Identifying Research Topics

The topic model technique Latent Dirichlet Allocation (LDA) [15] was used to answer Q2, Q3, and Q4. It is based on Bayesian models and is seen as a development of Probabilistic Latent Semantic Analysis [16,17].

A topic may be defined as a multinomial distribution of words in the vocabulary where each word has a different probability within each topic [18]. LDA is one of the unsupervised text mining methods, in which themes or topics of documents can be identified from a larger collection of compiled documents, called corpus. LDA adds a prior sparse Dirichlet distribution on items in a document, using sampling Gibb [19] to generatively assign the probabilities of the topics of each term, and then group the documents into their respective topics, assuming that the documents exhibit a combination of multiple subjects in different proportions. The goal of using LDA is to infer or estimate the latent variables, that is, to compute their conditional distribution documents. Equation (1) shows the statistical assumptions behind the LDA's generative process.

$$p(\beta_K, \theta_D, z_D, w_D) = \prod_{k=1}^{K} p(\beta_K|\eta) \prod_{m=1}^{M} p(\theta_m|\alpha) \prod_{n=1}^{N} p(z_{m,n}|\theta_m)P(w_{m,n}|z_{m,n}, \beta_{m,k}) \quad (1)$$

where $M$ denotes the number of documents, $N$ is number of words in a given document, and each topic $k$ is a multinomial distribution over the vocabulary and comes from a Dirichlet distribution $\beta_k \sim \text{Dir}(\eta)$, the Dirichlet parameter $\eta$ defines the smoothing of the words within topics, and $\alpha$ is the smoothing of the topics within documents. Every document is represented as a distribution over the topics and comes from a Dirichlet distribution $\theta_m \sim \text{Dir}(\alpha)$. The joint distribution of all the hidden variables, $\beta_K$ (topics), $\theta_M$ (document topic proportions within $M$), $z_M$ (word topic assignments), and observed variables $w_M$ (words in documents). The per-word topic assignment $z_{m,n}$, and the per-document topic distribution $\theta_m$, are the latent variables and are not observed. Moreover, the word $w_{m,n}$ depends on the per-word topic assignment $z_{m,n}$ and on all the topics $\beta_k$ (we retrieve the probability of $w_{m,n}$ (row) from $z_{m,n}$ (column) within the $K \times V$ topic matrix). We would have to condition on the only observed variable, that is the words within the documents, to infer the hidden structure with statistical inference. The conditional probability, also known as the posterior, is expressed by Equation (2).

$$p(\beta_K, \theta_M, z_M|w_M) = \frac{p(\beta_K, \theta_M, z_M, w_M)}{p(w_M)} \quad (2)$$

Although the posterior cannot be computed exactly due to the denominator [16], a close enough approximation to the true posterior can be achieved with statistical posterior inference. Mainly two types of inference techniques can be discerned: variational-based algorithms [20] and sampling-based algorithms [21]. An example of a sampling-based algorithm is the Gibbs sampler [22].

A simplified geometric interpretation of LDA is presented in Figure 1 considering only three words (w1,w2,w3) in the V-vocabulary and it is represented as a word simplex (V-dimensional). The word simplex is related to all the probability distribution of words. In addition, it can be seen how the topics, modeled as vocabulary distributions, are located within the simplex word (Figure 1). Figure 1 shows only three topics T, represented as a simplex topic of dimension (T-1). Thus, the documents modeled as distributions on the topics, are points on the simplex topic. For example document 1 would belong to topic 1; document 2 exhibits the same proportion in the three topics; while document 3 does not have proportions of topic 2.

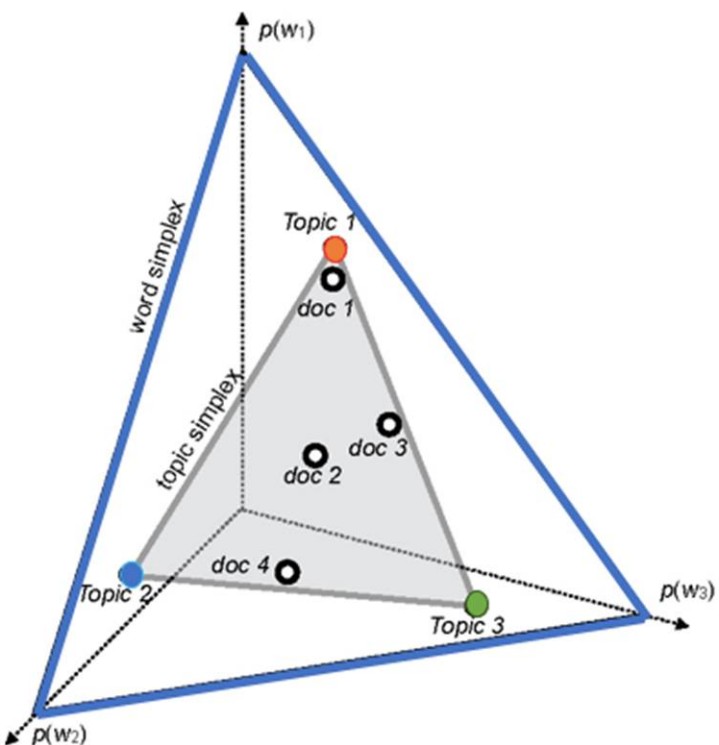

**Figure 1.** Geometric interpretation of LDA as a (V-1)-dimensional word simplex with V = w1, w2 and w3, with each point representing a discrete distribution of word probabilities. A point that is closer to one of the corners implies that the word has a higher probability mass. (Adapted from [18]).

### 2.3.1. Creation of LDA Model

LDA was used to extract meaningful information from the discovered articles. We combined the titles and abstracts of each article into a single variable. This variable was then used to serve as the text corpus for the entire data set.

Topic models are document latent variable models that leverage word correlations and latent semantic topics in a collection of texts [20]. This concept presupposes that the predicted number of topics k (i.e., latent variables) must be known in advance. Thus, the selection process of the right number of topics for a given collection of articles is not trivial. Simulations were carried out varying k from 4 to 30. 500 iterations were performed with the inference algorithm called Gibbs sampling [19]. A topic coherence metric [20] was used to estimate the quality of the LDA model. This is a measure of the human interpretability of a model of topics, and is believed to be a better indicator than computational metrics such as perplexity [23].

After determining the number of topics, we evaluated the most likely subject of each article and designated it as the article's primary topic.

### 2.3.2. Labeling Topics

Because algorithmic analyses are relatively restricted in their capacity to identify latent meanings of human language and the topics are not semantically labeled for the LDA model, manual labeling is regarded as a standard in topic modeling [24]. To provide a semantically correct interpretation, the topic was manually labeled by experienced clinicians and researchers independently using three sources of information: the most frequent word lists (most likely), a sample of the titles, and the abstracts of the five articles classified with the highest probability of belonging to a topic (Supplementary Materials, Table S1).

### 2.4. Quantitative Indices Used to Analyze the Trend of Topics

It is difficult to comprehend the subjects and trends intuitively due to the vast number of articles and hence the number of words. As a result, we employ certain quantitative

indicators given by Xiong et al. [25]. The indexes are described below. The distribution of topics over time is obtained by

$$\theta_k^y = \frac{\sum_{d \in m} \theta_{dk}}{n^m} \tag{3}$$

where $d \in m$ represents the articles published in a given month, $\theta_{dk}$ is the proportion of the $k$-th topic in each item and $n^m$ is the total number of articles published in the month [25].

Topic distribution across journals is defined as the ratio of the $k$-th topic in the journal

$$\theta_k^j = \frac{\sum_{d \in j} \theta_{dk}}{n^j} \tag{4}$$

where, $d \in j$ represents the articles in a particular journal, $\theta_{dk}$ the proportion of the $k$-th topic on each item, and $n^j$ is the total number of articles published in the journal $j$.

The proportion of the $k$-th topic in country c is defined as the topic distribution over countries, that is

$$\theta_k^c = \frac{\sum_{d \in c} \theta_{dk}}{n^c} \tag{5}$$

where $d \in c$ represents the articles in a specific country, $\theta_{dk}$ is the proportion of the $k$-th topic in each article, and $n^c$ is the total number of papers from the country $c$.

Topic distribution over time within a specific country, is defined as

$$\theta_k^{c,y} = \frac{\sum_{d \in c \cap d \in m} \theta_{dk}}{n^{c,m}} \tag{6}$$

where $d \in c \cap d \in m$ represents documents produced in a certain country over a certain month, $\theta_{dk}$ is the proportion of the $k$-th topic in each document, and $n^{c,m}$ the number of documents from country in month $m$.

We used simple regression slopes for each topic to facilitate the characterization of the topics in terms of their tendency [22]. The month was a dependent variable, and the proportion of the topics in the corresponding month was the response variable. The slopes derived by regression were positive or negative, and were classed as positive or negative trends, respectively. The statistical significance level was set at 0.01.

## 3. Results

### 3.1. Search Results

The initial database containing the documents retrieved after running the search query contained 161,421 documents; this sample was subjected to a filtering process in which repeated and poorly classified documents were eliminated, as well as those that did not contain a summary. There were a total of 126,334 papers in the final sample. Table 1 shows the summary produced, comprising basic statistics on the dataset studied.

A scientific production global map shows that COVID-19 research has been undertaken in all nations (excluding El Salvador, Central African Republic, South Sudan, Eritrea, Somaliland, Turkmenista, and the Democratic Republic of Korea) (Figure 2).

The top ten countries were the United States of America (26,814, 21.22%), China (11,375, 9.0%), Italy (7722, 6.11%) percent), United Kingdom (7522, 5.95 % percent), India (6726, 5.32%), Canada (3591, 2.84%), Spain (3465, 2.74%), Germany (3129, 2.48%), France (3129, 2.48%) and Iran (2843, 2.25%).

The results show that the articles published during the period between February 2020 and January 2022, experienced a compound monthly growth rate close to 34.6% (from 101 to 126,334) (Table 2).

In terms of sources (of the 7040 registered), the International Journal of Environmental Research and Public Health, PLoS ONE, and Scientific Reports have published the largest number of articles on COVID-19, having collectively published close to 5% of all publications on COVID-19 in the study period (Table 3).

**Table 1.** Main statistics about the COVID-19 collection.

| Description | | Result |
|---|---|---|
| Main information about data | Timespan | February 2020: January 2022 |
| | Sources | 7040 |
| | Documents | 126,334 |
| | Average years from publication | 1.46 |
| Document contents | Keywords plus (id) | 13,001 |
| | Author's keywords (de) | 112,867 |
| Authors | Authors | 440,259 |
| | Author appearances | 960,863 |
| | Authors of single-authored Documents | 5374 |
| | Authors of multi-authored Documents | 434,885 |
| Authors collaboration | Single-authored documents | 6698 |
| | Documents per author | 0.287 |
| | Authors per document | 3.48 |
| | Co-authors per documents | 7.61 |
| | Collaboration index | 3.64 |

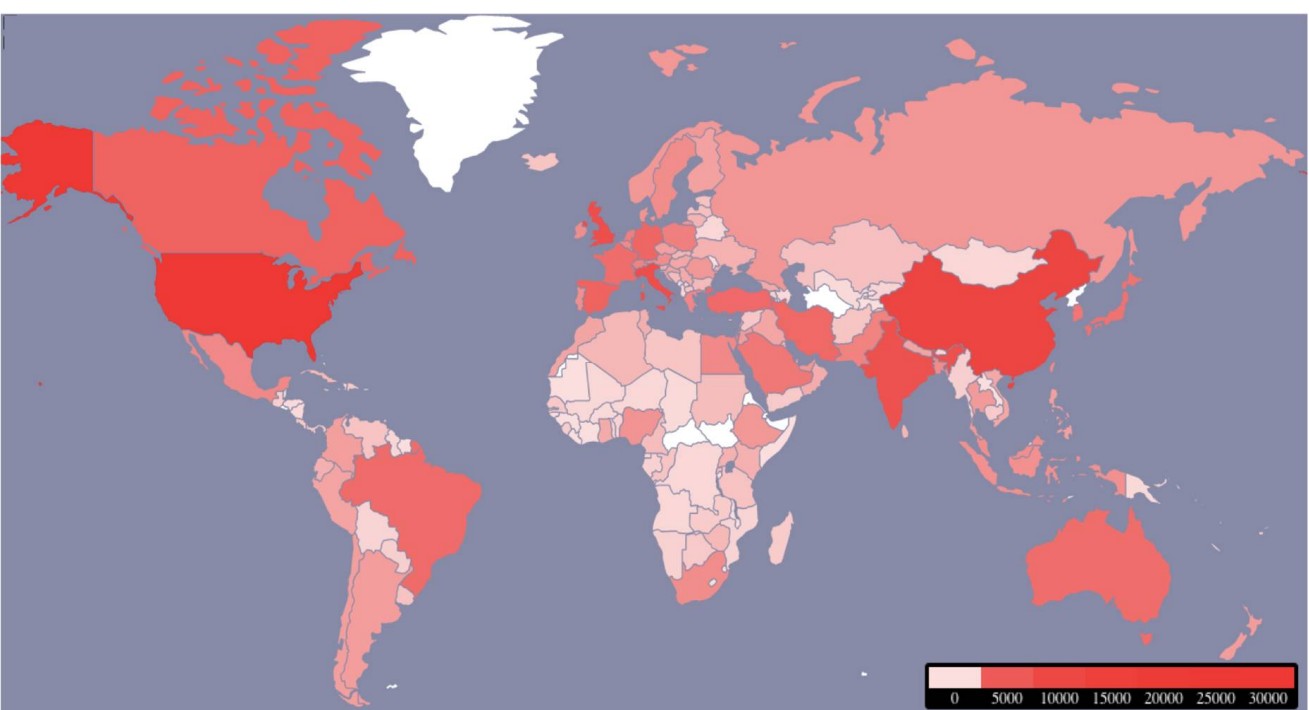

**Figure 2.** Geographical origin distribution of the 126,334 articles published on COVID-19 analyzed.

*3.2. LDA Modeling and Topics*

The LDA model with the highest coherence contains 16 topics. Table 4 shows for each of them the 15 most common terms, the label, and the number of published articles referring to them. The topics with the highest number of articles were: t_16 (Political and health factors), t_13 (Mental health), and t_15 (Etiopathogenesis), while the t_9 (Information synthesis methods) had the lowest number of articles.

### 3.2.1. Trend of Topics

The trend of each of the 16 topics over time was discovered. It can be observed that the probabilities of eight of them gradually increased over time (t_2 Prevention, t_3 Telemedicine, t_4 Vaccine immunity, t_5 Machine learning, t_7 Academic parameters, t_8, Risk factors and morbidity and mortality, t_9 Information synthesis methods, and t_13 Mental health), in five of them the probability decreased (t_6 Epidemiology, t_10 COVID-19 pathology complications, t_12 Diagnostic test, t_15 Etiopathogenesis, and t_16 Political and health factors), while the remainder fluctuated over time (t_1 Therapeutics, t_11 Pharmacological and therapeutic target, and t_14 Repercussion health services), without prominent trends (Figure 3).

**Table 2.** Main statistics about the COVID-19 collection.

| Month | Year | Number | Accumulated |
|---|---|---|---|
| February | 2020 | 101 | 101 |
| March | 2020 | 558 | 659 |
| April | 2020 | 2082 | 2741 |
| May | 2020 | 3476 | 6217 |
| June | 2020 | 4255 | 10,472 |
| July | 2020 | 4685 | 15,157 |
| August | 2020 | 4307 | 19,464 |
| September | 2020 | 4819 | 24,283 |
| October | 2020 | 5193 | 29,476 |
| November | 2020 | 4765 | 34,241 |
| December | 2020 | 4718 | 38,959 |
| January | 2021 | 17,640 | 56,599 |
| February | 2021 | 6345 | 62,944 |
| March | 2021 | 5984 | 68,928 |
| April | 2021 | 5421 | 74,349 |
| May | 2021 | 5578 | 79,927 |
| June | 2021 | 5803 | 85,730 |
| July | 2021 | 6121 | 91,851 |
| August | 2021 | 5529 | 97,380 |
| September | 2021 | 5736 | 103,116 |
| October | 2021 | 5880 | 108,996 |
| November | 2021 | 5546 | 114,542 |
| December | 2021 | 5593 | 120,135 |
| January | 2022 | 6199 | 126,334 |

**Table 3.** Top 10 most important sources in terms of number of publications.

| Source | Abbreviation | *n* | (%) |
|---|---|---|---|
| International Journal of Environmental Research and Public Health | *Int. J. Environ. Res. Public Health* | 3304 | 2.62 |
| PLoS ONE | *PLoS ONE* | 2057 | 1.63 |
| Scientific Reports | *Sci. Rep.* | 1348 | 1.07 |
| Frontiers in Psychology | *Front. Psychol.* | 997 | 0.79 |
| BMJ Open | *BMJ Open* | 923 | 0.73 |
| Journal of Clinical Medicine | *J. Clin. Med.* | 900 | 0.71 |
| Journal of Medical Virology | *J. Med. Virol.* | 865 | 0.68 |
| Cureus | *Cureus* | 817 | 0.65 |
| Frontiers in Public Health | *Front. Public Health* | 813 | 0.64 |
| International Journal of Infectious Diseases | *Int. J. Infect. Dis.* | 786 | 0.62 |

### 3.2.2. Topic Distributions of Various Journals

In Figure 3, we depict the topic distribution of journals as a heatmap, with the intensity of the pixel representing the probability that a given topic is mentioned in a certain journal. Although the content of many of the journals included in our study overlaps to some

extent, it is feasible to identify journals that have relatively wide scopes, while others appear to specialize in certain topics. For instance, the journals Frontiers In Psychology and Frontiers In Psychiatry focus on the topic t_13 (Mental health). In addition, we performed a hierarchical cluster analysis on the contents of the selected journals by computing the Euclidean distance between each pair of journals. Dendrogram is shown on the left panel of Figure 4, where journals were classified into seven groups. Two of the 30 journals considered in the analysis formed the isolated cluster 6 (Vaccines) and cluster 7 (BMJ Case Rep.) while the remaining journals can be classified into five groups.

**Table 4.** 16 topics discovered from 126,334 articles published on COVID-19 in the period February 2020–January 2022. Each topic shows the 15 most likely terms (that is, the words with the highest probability), the label, and the number of published articles belonging to each topic.

| Topic | Label | Top_terms | Articles *n* (%) |
|---|---|---|---|
| t_1 | Therapeutics | treatment, trial, clinic, group, therapi, control, drug, effect, treat, clinic_trial, dose, efficaci, receiv, improv, random | 3671 (2.91) |
| t_2 | Prevention | survei, worker, particip, health, risk, healthcar, associ, prevent, cross, pandem, section, cross_section, factor, behavior, protect | 6380 (5.05) |
| t_3 | Telemedicine | servic, women, pandem, clinic, provid, telemedicin, visit, telehealth, health, pregnant, access, deliveri, person, consult, medic | 4857 (3.84) |
| t_4 | Vaccine inmunity | vaccin, antibodi, immun, respons, dose, igg, neutral, infect, anti, effect, hesit, mrna, individu, receiv, level | 4146 (3.28) |
| t_5 | Machine learning | model, base, predict, method, data, perform, propos, mask, develop, learn, imag, system, valid, time, detect | 6781 (5.37) |
| t_6 | Epidemiology | case, infect, countri, data, rate, number, transmiss, model, death, popul, spread, measur, epidem, diseas, outbreak | 10,784 (8.54) |
| t_7 | Academic parameters | student, pandem, educ, nurs, onlin, learn, medic, experi, social, train, school, particip, resid, program, media | 7693 (6.09) |
| t_8 | Risk factors and morbidity and mortality | mortal, risk, associ, sever, outcom, diseas, hospit, icu, higher, admiss, factor, death, cohort, group, clinic | 10,665 (8.44) |
| t_9 | Information synthesis methods | review, systemat, search, analysi, systemat_review, includ, meta, literatur, report, meta_analysi, databas, evid, data, pubm, identifi | 1955 (1.55) |
| t_10 | COVID-19 pathology complications | symptom, case, diseas, sever, clinic, report, infect, ct, children, present, group, find, pneumonia, includ, acut | 7655 (6.06) |
| t_11 | Pharmacological and therapeutic target | protein, drug, viral, human, viru, cell, target, bind, ac, spike, infect, potenti, activ, genom, variant | 7467 (5.91) |
| t_12 | Diagnostic test | test, posit, detect, pcr, sampl, infect, rt, neg, rt_pcr, assai, sensit, viral, diagnost, respiratori, swab | 5635 (4.46) |
| t_13 | Mental health | pandem, health, mental, anxieti, mental_health, stress, depress, psycholog, symptom, associ, social, impact, level, particip, increas | 12,236 (9.69) |
| t_14 | Repercusion health services | pandem, period, cancer, surgeri, compar, lockdown, impact, increas, emerg, time, surgic, decreas, number, chang, march | 6412 (5.08) |
| t_15 | Etiopathogenesis | infect, diseas, sever, respiratori, syndrom, acut, cell, acut_respiratori, immun, respiratori_syndrom, sever_acut, respons, system, inflammatori, associ | 12,080 (9.56) |
| t_16 | Political and health factors | health, pandem, public, system, manag, challeng, respons, global, commun, diseas, develop, emerg, public_health, provid, impact | 17,917 (14.18) |

### 3.2.3. Topic Distribution over Country

Following the methodology used in the analysis of journals, in Figure 5, we can see a heatmap with a dendrogram in the left panel. We only considered 35 countries for the analysis, of which 30 are considered leaders in the field of scientific research based on their publication volume according to the Nature Index [26]. In general, topics t_9 (Information synthesis methods), t_3 (Telemedicine) and t_1 (Therapeutics) were the ones that generated less interest from the countries evaluated, while t_16 (Political and health factors) was the most prevalent in South Africa, Australia, Ireland, Canada, Singapore, United Kingdom and United States of America (USA).

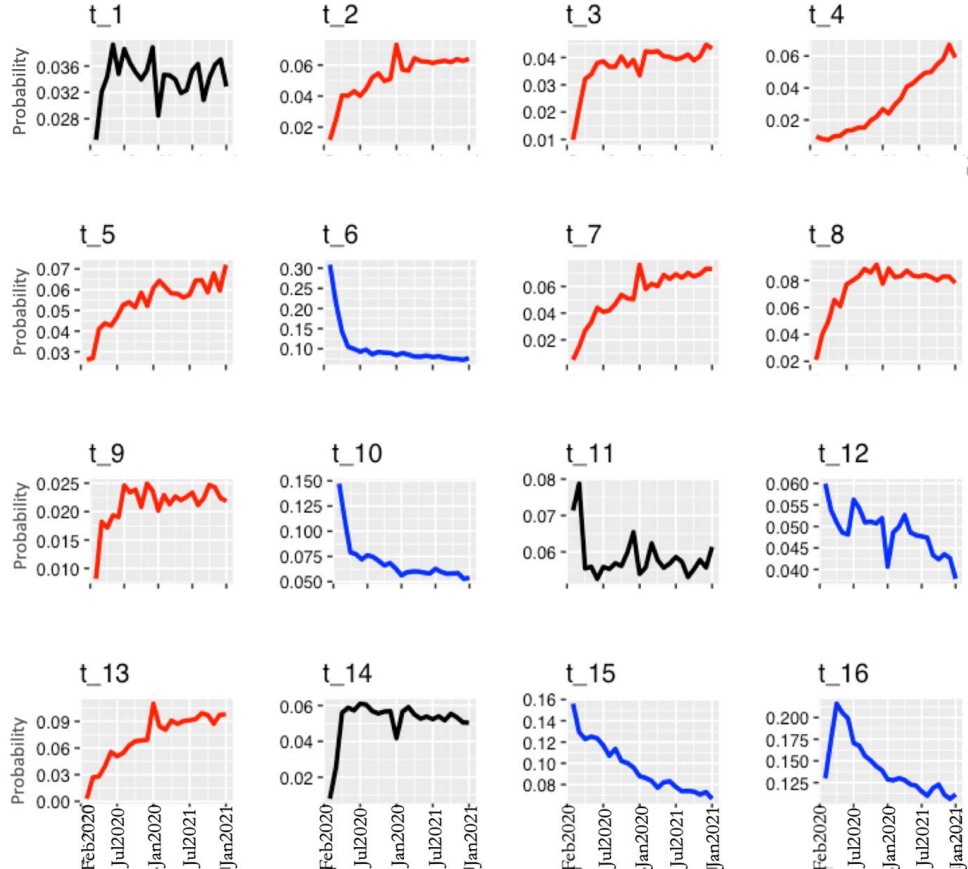

**Figure 3.** Topic trends research in COVID-19 during February 2020-January 2022. The red color indicates topics with increasing tendency, blue with decreasing tendency, and black fluctuating.

We also investigated the distribution of topics by country over time to determine how topics changed in various countries over time.

In general, t_4 (Vaccine inmunity) was the topic that showed a positive trend in all the countries (except Ireland) considered in the analysis, while t_15 (Etiopathogenesis) and t_16 (Political and health factors) showed a negative or fluctuating trend in the countries analyzed (Table 5).

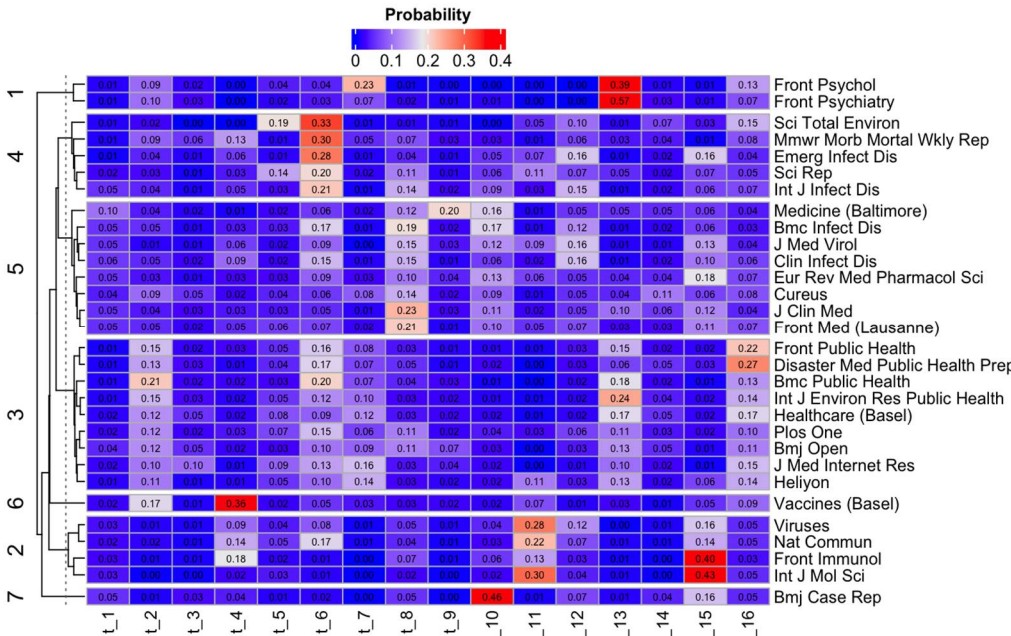

**Figure 4.** Heatmap overview of the proportional topic in the top-30 analyzed journals. Values are in percentages and row totals sum up to 100%.

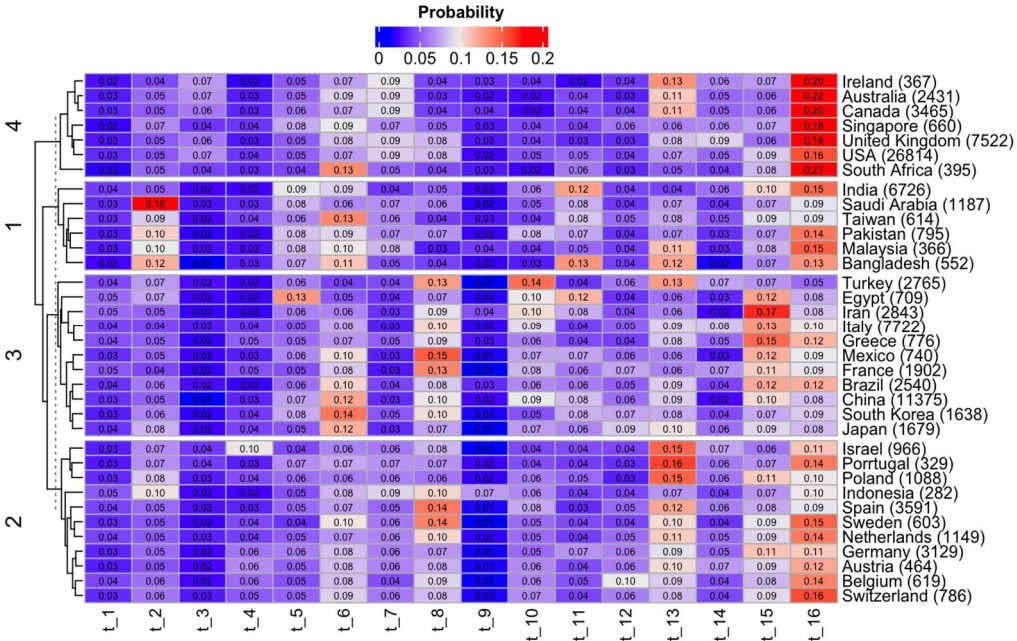

**Figure 5.** Heatmap overview of the proportional topic in the top-35 analyzed countries. Values are in percentages and row totals sum up to 100%. In parentheses, the number of articles is shown.

**Table 5.** Topic trends research in COVID-19 during February 2020–January 2022. Red color indicates increasing tendency, blue decreasing tendency, and white fluctuating or no prominent trends.

| Country | t_1 | t_2 | t_3 | t_4 | t_5 | t_6 | t_7 | t_8 | t_9 | t_10 | t_11 | t_12 | t_13 | t_14 | t_15 | t_16 |
|---|---|---|---|---|---|---|---|---|---|---|---|---|---|---|---|---|
| USA | | R | R | R | R | B | R | R | R | B | R | B | R | | B | B |
| China | | R | B | R | R | B | R | | | B | R | B | | | B | |
| Italy | | R | | R | R | B | R | | | | | | R | | B | |
| United Kingdom | R | R | R | R | R | B | | | | R | R | | R | | | B |
| India | | R | | R | R | B | R | R | | R | R | | | | | B |
| Spain | | R | | R | R | B | R | R | | | | | R | | | B |
| Canada | | R | | R | | B | | R | | | | B | R | | | B |
| Germany | | R | | R | R | | | R | | | | B | | | | B |
| Iran | | | | R | R | | R | R | | | | | R | | | B |
| Turkey | R | R | | R | R | B | R | R | R | | | | R | | | B |
| Brazil | | | R | R | R | B | R | | | | | | R | | B | B |
| Australia | | | R | R | R | B | | | | | | | | | B | B |
| France | | R | R | R | R | | R | | | B | | | R | | B | B |
| Japan | | R | R | R | R | B | R | | | B | | | R | | | B |
| South Korea | | R | | R | R | B | R | | | B | | B | R | | | |
| Saudi Arabia | | R | | R | R | B | | | | | | | | | B | B |
| Netherlands | | | | R | R | | | | | | | | R | R | B | |
| Poland | | | | R | R | R | R | | | | | | R | | B | |
| Israel | | | | R | R | | R | R | | | | | | | B | |
| Pakistan | | | | R | R | | R | R | | | | | R | R | B | |
| Switzerland | | R | | R | R | | R | | | | | | R | | B | |
| Greece | | | | R | R | B | | | | | | | R | | B | |
| Mexico | R | R | | R | R | B | | R | | | B | | R | | | B |
| Egypt | | | | R | R | B | R | R | | | | | R | | B | |
| Singapore | | R | | R | R | B | R | | | | | | R | | B | B |
| Belgium | | R | | R | R | | R | | | B | | | R | | B | B |
| Taiwan | | | | R | R | | R | | | | | | R | | B | B |
| Sweden | R | R | | R | R | B | R | | | | | B | R | R | | |
| Bangladesh | | R | R | R | R | B | R | | | | R | R | R | | B | |
| Austria | | | | R | R | | R | R | | | | | | | | |
| South Africa | | | | R | R | | R | R | | | | | R | | | |
| Ireland | | R | | R | R | | R | R | | | | | R | | | |
| Malaysia | | R | | R | R | B | | | | | | | R | | B | |
| Portugal | | | | R | R | B | | | | | | | R | | B | |
| Indonesia | | | R | R | R | | | | | | | | | | B | |

## 4. Discussion

The rapid increase in publications related to COVID-19 is unprecedented in the scientific literature, even compared to the Zika virus outbreak in Latin America (January 2016), when the WHO declared a health emergency of international concern [27]. In this case, there were only 644 publications on PubMed for the first six months after the declaration, which highlights the big difference with the 15,557 publications on COVID-19 between February and July 2020. Another pertinent comparison can be made with the global pandemic caused by influenza A (H1N1), first detected in North America in 2009 [28]. In fact, while the first publication of clinical trials on COVID-19 was made 44 days after the declaration of a pandemic by the WHO [29], for H1N1, this occurred 190 days after the declaration [28]. However, not only the number of articles published was exceptional, but also the period of time between data collection and publication of the articles was surprising. This faster publication procedure was largely made possible by a shorter peer review process. Horbach [30] evidenced this in his study with the peer-review process of 14 medical publications. In fact, journal processing time was lowered by 49%. Researchers on topics related to COVID 19 have worked beyond their means, both researching and reviewing the literature, while it seems reasonable that journals might find it difficult to attract reviewers with relevant experience, as they are likely to be active scientists, it

seems that journals are finding enough reviewers willing to review articles related to the coronavirus in a very short time [31].

Wanting to share information quickly has often led to a decrease in the evaluation time of articles and more lax reviews, having accepted articles of lower quality, prioritizing immediacy in information over quality [32]

The above confirms the growing public and scientific interest, given the fact that the disease represents a major threat to public health worldwide, but also to the economic and social consequences associated with it. Therefore, it is not surprising that COVID-19 research has seen an unprecedented increase since the beginning of the pandemic [33].

The results also suggest that the USA exceeds countries such as China, Italy, the United Kingdom, India, Canada, Spain, Germany, France, and Iran in number of articles published. This fact is not surprising given the amount of USA government funds that were invested in COVID-19 research [33]. Publications from other geographic areas are substantially less abundant, with gaps particularly visible in Africa, Latin America, Eastern Europe, and Central Asia.

Some systematic reviews have been published on COVID-19, these require a lot of research time and have generally focused on specific aspects of the pandemic [4–6]. Those works also analyzed reports on COVID-19 in the media [34], social networks such as Twitter [35], and Sina-Weibo (a Twitter system used in China) [36].

Unlike the aforementioned reviews, this study did not focus on specific aspects of the pandemic, but instead reviewed all the scientific literature related to COVID-19 during the two years after the pandemic was declared. In particular, LDA allowed for the evaluation of the variation of the research in the medium term. This technique also offers the possibility to conduct a more in-depth analysis on a particular topic identified. We identified 16 topics (namely, t_1 Therapeutics, t_2 Prevention; t_3 Telemedicine, t_4 Vaccine immunity, t_5 Machine learning t_6 Epidemiology, t_7 Academic parameters, t_8 Risk factors and morbidity and mortality, t_9 Information synthesis methods, t_10 COVID-19 pathology complications, t_11 Pharmacological and therapeutic target, t_12 Diagnostic test, t_13, Mental health, t_14 Repercusion health services, t_15 Etiopathogenesis and t_16 Political and health factors) it was possible to categorize the scientific papers on COVID-19 that were published during the first two years of the pandemic.

Älgå et al. [2] explored the scientific literature on COVID-19 (16,670 articles, using PubMed as in our study) in the time period between February and June 2020 using LDA. In this case, 14 topics were identified (namely, Therapies and vaccines, Risk factors, Health care response, Epidemiology, Disease transmission, Impact on health care practices, Radiology, Epidemiological modeling, Clinical manifestations, Protective measures, Immunology, Pregnancy, and Psychological impact). Therefore, it was observed that some of the topics coincide with some labeled in this study. However, there were differences regarding the most prevalent topics. While [2] reported that the most prevalent topics were health care response, clinical manifestations, and psychological impact, in our case they were t_16 Political and health factors, t_13, Mental health, and t_15 Etiopathogenesis. These differences can be explained by the time period evaluated. Furthermore, since the COVID-19 epidemic is still ongoing, the topics of study will most likely continue to change over time.

Among the rising academic attempts to address COVID-19 problems, a large portion of the research has naturally concentrated on elements relating to Political and health factors, Epidemiology and Risk factors, morbidity and mortality, Mental health, and, Etiopathogenesis.

It should be noted that the study was constrained by the exclusion of grey literature, books, book chapters, reviews, and reports. The data was acquired entirely from the PubMed database and only scientific articles were considered. Academics may opt to conduct future research using other databases, such as Scopus and Web of Science, which include non-indexed journals not included in PubMed. In this sense, future research might compare the findings of this study to those obtained from other databases.

In sum, the findings of this study may be used to illustrate how the medical research community reacts and what issues are prioritized. On the other hand, it was easy to identify how research efforts were distributed globally and how they changed over time.

## 5. Conclusions

Scientific research and data play a very important role in the early control and prevention of disease outbreaks and epidemics. It is of great interest to quickly share all information with the public, researchers, government organizations, and institutes, both nationally and internationally. An example of this was the surprising amount of studies on COVID-19 that have been published since the novel coronavirus was originally identified.

In this work, the variations in the COVID-19 study that were available over the first two years of the pandemic were highlighted. Therefore, this study demonstrated that the United States of America, China, and Italy have leading roles in COVID-19 research. In addition, through LDA modeling, a list of 16 topics was obtained and important temporal trends could be identified.

In sum, the outcomes can provide new study guidelines, as well as aid in understanding research trends, in the context of worldwide occurrences, useful for academics and policymakers. Furthermore, the results achieved showed that topic modeling is a quick and efficient way to evaluate the progress of a huge and quickly developing a research topic, such as COVID-19. Additionally, and perhaps even more importantly, the methodology used has the potential to identify topics for future research, not only in studies on pandemics but also as a tool for the identification and review of scientific literature in other fields which may be of great public interest.

**Supplementary Materials:** The following supporting information can be downloaded at: https://www.mdpi.com/article/10.3390/computation10090156/s1, Table S1: Five-top papers for each estimated topic, based on topic-document probability matrix.

**Author Contributions:** Conceptualization J.D.L.H.-M., S.M. and M.J.F.-G.; methodology, J.D.L.H.-M. and S.M.; software J.D.L.H.-M.; validation, Y.G.S., and M.J.F.-G.; formal analysis, J.D.L.H.-M., S.M. and M.J.F.-G.; investigation, J.D.L.H.-M. and Y.G.S.; data curation, J.D.L.H.-M.; writing—original draft preparation, J.D.L.H.-M.; writing—review and editing, S.M., Y.G.S and M.J.F.-G.; visualization, J.D.L.H.-M.; supervision, M.J.F.-G. and S.M.; project administration, S.M. and M.J.F.-G. All authors have read and agreed to the published version of the manuscript.

**Funding:** This research received no external funding.

**Institutional Review Board Statement:** Not applicable.

**Informed Consent Statement:** Not applicable.

**Data Availability Statement:** Not applicable.

**Acknowledgments:** Regarding Susana Mendes, this work was funded by national funds through FCT - Fundação para a Ciência e a Tecnologia, I.P., under the project MARE (UIDB/04292/2020 and UIDP/04292/2020) and the project LA/P/0069/2020 granted to the Associate Laboratory ARNET.

**Conflicts of Interest:** The authors declare no conflict of interest.

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
