# Peer review of "Capturing the Complexity of COVID-19 Research: Trend Analysis in the First Two Years of the Pandemic Using a Bayesian Probabilistic Model and Machine Learning Tools"

_computation, doi:10.3390/computation10090156_

Round 1

Reviewer 1 Report

The proposed manuscript is devoted to a study of the main topics of the scientific literature on COVID-19 that has been published since its inception and their evolution and distribution within the period between February 2020 and January 2022. The search of the authors was carried out in PubMed extracting topics by the use of text mining and latent Dirichlet allocation modeling. Further, the authors performed a trend analysis of the temporal research variations for each topic. In addition, they studied the distribution of the main topics between countries and journals.

Preliminaries to the research area are provided. In particular, they briefly describe the events related to the coronavirus outbreak and the appearance of high number of scientific publications concerned with the pandemic. They make the conclusion of the necessity of this amount of literature to be systematized and classified.

The authors formulate the main questions related to their study. They describe in detail their data collection, its preprocessing, and the model technique used to study the main research problems. They present clearly the obtained results and provide a thorough discussion.

The presentation of the main results is clear and comprehensive. From a formal point of view, all the contents seem to be correct. The results are valuable and worthy of being published taking into account their possible applications in the field of research related to COVID-19.

My previous remarks have been taken into account.

Additional minor revisions are suggested to improve the quality of the exposition:

p. 1 line 25: A comma should be inserted between “United States of America” and “China” as well as after “India” instead of full stop before the word “respectively”, which should end the sentence.

p. 3 line 95: I suggest to write ”Data preprocessing was carried out using the web-based tool LDAShiny [15], a package to R programming language [13].” instead of ”Data preprocessing was carried out using LDAShiny [15] (a web-based tool), a package to R programming language [13].”

Author Response

The authors agree and appreciate the comment. Suggested changes were made.

  1. 1 line 25: A comma should be inserted between “United States of America” and “China” as well as after “India” instead of full stop before the word “respectively”, which should end the sentence.

...United States of America, China, Italy, United Kingdom, and India, respectively

  1. 3 line 95: I suggest to write ”Data preprocessing was carried out using the web-based tool LDAShiny [15], a package to R programming language [13].” instead of ”Data preprocessing was carried out using LDAShiny [15] (a web-based tool), a package to R programming language [13].

Data preprocessing was carried out using the web-based tool LDAShiny [15], a package to R programming language [13]…

Reviewer 2 Report

I cannot see the response to my comments.

Goingf over my previous comments, and highlights of a new manuscipt, it is clear that authors did not bother give any attention to the paper. 

Starting from the first line of manuscript, "Publications about COVID-19 have occurred practically since the first out-break. " So what? what does it mean and how it help?

Go over the whole paper and highlight the changes and respond to my previous comments. 

Author Response

I cannot see the response to my comments.

The authors agree and understand their comments, in this regard the conclusions section was added in response to their suggestions and all their recommendations and corrections were taken into account.

Goingf over my previous comments, and highlights of a new manuscipt, it is clear that authors did not bother give any attention to the paper. 

Starting from the first line of manuscript, "Publications about COVID-19 have occurred practically since the first out-break. " So what? what does it mean and how it help?

We agree with your assessment and now the text is as follows:

L.15 Publications about COVID-19 have occurred practically since the first out-break. Therefore, stud-ying the evolution of the scientific publications on COVID-19 can provide us with information on current research trends and can help researchers and policymakers to form a structured view of the existing evidence base of COVID-19 and provide new research directions

Go over the whole paper and highlight the changes and respond to my previous comments. 

Reviewer 3 Report

Referee Report on

Capturing the Complexity of Covid-19 Research: Trend Analysis in the First Two Years of the Pandemic Using a Bayesian Probabilistic Model and Machine Learning Tools

This paper is well prepared, written and organized, it presents a scientific literature study on COVID-19 that has been published since its inception and to map the evolution of research in the time range between February 2020 and January 2022. The search was carried out in PubMed extracting topics using text mining and latent Dirichlet allocation modeling and performed a trend analysis to analyze the temporal variations in research for each topic. It also proposes a study the distribution of these topics between countries and journals. 126,334 peer-reviewed articles and 16 research topics were identified.

I recommend accepting this paper after minor revisions:

1.   Moderate English changes required.

2.   Into page 9 line 254, it should be "Table 4 shows" instead of "Table 3 shows".

Author Response

Reviewer 3

The authors agree and appreciate the comment. Suggested changes were made.

I recommend accepting this paper after minor revisions:

  1. Moderate English changes required.

The authors agree and took this into account in the review.

  1. page 9 line 254, it should be "Table 4 shows" instead of "Table 3 shows".

Correction was made

Round 2

Reviewer 2 Report

addressed